# Post-prediction confidence training complements supervised learning

## Abstract

Wrong prediction is bad. For users, having high confidence on a wrong prediction is even worse. Since even the best-trained class-label predictor will have some chance of making mistakes, users, especially in some AI application areas such as personalized medicine, may want to tell the high quality predictions from the low quality ones. In convolutional neural networks (CNN), confidence on a prediction is associated with the softmax output layer, which gives a probability distribution on the class-labels. But even a prediction with 95% probability concentrated on one class may still turn out wrong many times more often than the anticipated rate of 5%. There are at least three main sources of uncertainty to cause a large anticipation gap. The first one is that some of the test samples may not belong to the same distribution of the training samples. The second one is the sever population heterogeneity within each class, causing the variation of prediction quality across some hidden subpopulations. The third one is the imperfectness of the prediction model. While most researches are focused on the first source of prediction uncertainty, the other two receive much less attention. Here we take a different approach, termed post-prediction confidence training (PPCT), to guide users how to discern the high-quality predictions from the low-quality ones. Distinctively different from other methods including conformal prediction, PPCT entertains all three sources of uncertainty by searching features to anchor the criticism of prediction quality. An enhancement to CNN configuration is required during network training. We propose a blueprint by coupling each logit node (T channel) in the layer feeding to softmax with an additional node (C channel) and using maxout to link the pair to the softmax layer. The C channel is introduced to counter the T channel as a contrastive feature against the feature of the target class. A high-quality prediction must follow a logically-lucid pattern between T and C for every class. Successful implementation of our methods on popular image datasets are reported.

## 1 Introduction

Despite the enormous accomplishments in supervised learning, the research is thin on a post-prediction confidence issue concerning how trustworthy a given prediction is. This concern is not about the overall performance of a prediction algorithm. Rather it is about the particular prediction the algorithm is made for an input that the user is interested in. A great mindset difference is clear between the machine learning developers and users. In developing algorithms for supervised learning, the overall prediction-accuracy improvement is the developer's primary task. In contrast, an individual user may worry more about whether a specific prediction made by the algorithm is trustworthy or not. An algorithm with demonstrated 99% or higher overall accuracy rate is no guarantee to ease the user's anxiety that the prediction for his/her case may go wrong. Such demands can be specially keen in some AI applications for the personalized usage such as medical imaging or other health-related recommendations. At the bottom line, users would like to be informed not just what the prediction is but also whether it is a high-quality prediction or not. Up to date, not enough search attention has been paid to meet the demand on post-prediction confidence evaluation. Without well-disciplined rule guidance, discerning a high-quality prediction from a low-quality one is often decided on an ad hoc basis.

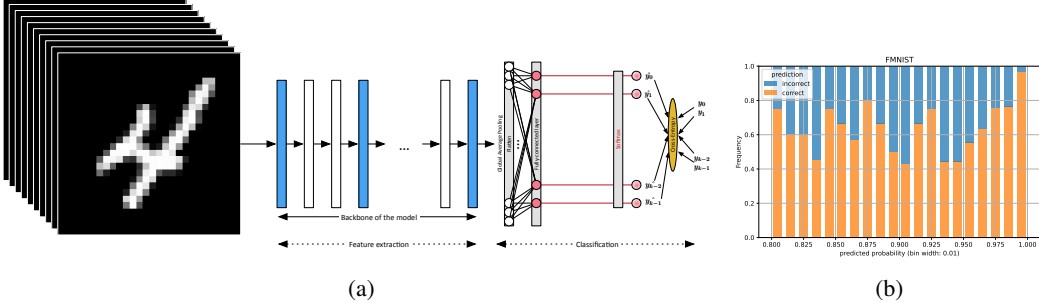

(a)                                                              (b)

Figure 1: (a) An illustration of the architecture of a conventional neural network, (b) The reliability diagram.

In this paper, we focus on the deep neural network for object-class membership prediction. The configuration of a deep convolutional neural network (CNN) can be broken down into two parts - the first part for image feature representation in a d-dimension space and the second part for implementing multi-class logistic regression with d regressors. Formally, the model can be written as $\hat{\mathbf{y}} = \text{softmax}(\mathbf{t})$; $\mathbf{t} = \phi\mathbf{h}$; $\mathbf{h} = f(\mathbf{x}, \theta)$, where $\mathbf{x}$ is the input image, $\mathbf{h}$ is a d-dimensional image feature vector, $f$ denotes a nonlinear transformation constituted of many layers of nodes with the connection weight $\theta$, $\phi$ is a K by d matrix and $\mathbf{t}$ is a vector of K target-class features (the logit layer of K-class logistic regression). The predicted class is $\hat{k} = \arg\max\{\hat{y}_k : k = 1, \cdots, K\}$. The task of network parameter training is to optimize the overall prediction accuracy. In multi-class logistic regression, $\hat{\mathbf{y}}$ is interpreted as an estimate of the class-probability distribution with input regressor $\mathbf{h}$, that is $p_k = P\{y_k = 1|\mathbf{h}\}$ for $k = 1, \cdots, K$. Thus seeing a prediction on $\mathbf{x}$ with very high probability, $\hat{y}_{\hat{k}}$, higher than 0.95, for example, one would have a high anticipation that the prediction is true. However, over-reliance on this quantity can be disappointing. As an illustration, we run a deep CNN on the image dataset FMINST (Xiao et al. (2017)). Among all test cases falling between .95 and .96, less than 60% cases are correctly predicted. Compared to the anticipated error rate 5%, the actual error rate is more than 8 times higher. For this experiment, the overall accuracy is 95% (Figure 1(b)).

Researchers have long been aware of the weakness of making a single prediction without attaching any level of confidence. Several approaches to quantify confidence have been developed for general applications in supervised learning. Because difference in the population distribution between the training dataset and testing dataset is a likely reason for a disappointing prediction, several p-value based confidence measures developed under the paradigm of conformal prediction are proposed to qualify the likelihood that the test case may come from a different distribution. However, to the best of our knowledge, successful applications of these methods, including inductive conformal prediction, to deep CNN on large image datasets have not been reported.

Setting aside the computational burden issue, there are two more major sources of prediction uncertainty that most confidence measures have not yet paid the due attention to: population heterogeneity and prediction model imperfectness. For the datasets that deep CNN are designed for, we expect that each class is composed of many subpopulations. Due to model imperfectness, the prediction performance is likely to vary substantially between subpopulations. Therefore it is likely that a disappointing prediction may come from some particular subpopulations that model imperfectness is more pronounced. Here is a summary of our contributions :

**1.** We pinpoint three sources that may cause prediction uncertainty in deep CNN: distributional non-conformability, population heterogeneity, model imperfectness.

**2.** To address the issue of model inadequacy in deep CNN due to population heterogeneity, we analyze the difficulties in conducting model criticism for large models involving more parameters than the sample size of training set. Leveraging the decomposability of the CNN configuration, we conceive a strategy to focus on the component of multi-class logistic regression.

**3.** We propose a post-prediction confidence training (PPCT) task to complement the prediction-accuracy task in supervised learning. PPCT aims at finding features to anchor the criticism of prediction quality due to the aforementioned three sources of uncertainty.

**4.** Keeping the same blueprint of network configuration for testing, our strategy is to modify the blueprint for training by augmenting the logit layer with a dual-channel representation layer that couples each target-class feature node (T) with a contrastive feature node (C). C versus T is interpreted as a duel between positive feature and negative feature to compete for representing the class. Thus in addition to the target features, additional features hidden in the image feature layer, can be brought up for challenging the representativeness of the target-class features.

**5.** We theorize a set of rules to differentiate the roles of T node and C node. Following these rules, a molding plate, named Yo-Yo, is introduced to represent the ideal pattern of features captured through the two channels after training. Algorithms are developed to facilitate PPCT.

**6.** Our method outputs a summary of post-prediction diagnostics, ready for users to evaluate the prediction quality from multiple angles. In addition to the categorical and the finer-numerical grading, plots for visualizing the C-T scoring patterns are also provided.

## 2 RELATED WORK

Several lines of advances in machine learning have influence our work. Although different, our idea of bringing in a contrastive feature for prediction quality criticism may find roots in the contrastive self-supervised learning based on joint-embedding with a siamese architecture (Bromley et al., 1993)), and the the notion of "contrastive between samples or between dimensions" (Garrido et al. (2023)).

The use of maxout operator of is critical in our network configuration for training. Leveraging the property of being an universal approximator, the maxout network configuration was designed as an alternative to the technique of 'drop-out' model averaging for facilitating optimization by dropout and improving prediction accuracy (Goodfellow et al. (2013)). Our use of maxout is not aimed at prediction accuracy improvement, however.

Conformal prediction (CP) is invented by Vovk et al. (2005) as a method to counter the Bayesian methods for obtaining confidence values without knowing the prior probabilities. The key notion is a p-value reflecting the unusualness of a candidate prediction value compared to the training sets. Since its inception, CP has stimulated many new ideas, techniques and theory covering regression, classification and clustering (Vovk et al., 2009; Lei et al., 2015; 2018).

CP is computational inefficient, especially for large neural network. Using the idea of split sample, inductive CP (ICP) is first introduced for ridge regression (Papadopoulos et al. (2002)). Papadopoulos et al. (2007) demonstrated the successful implementation of ICP for neural networks without the massive computation overhead of CP. Matiz & Barner (2019) applied IPC to active learning. Unlike our approach, the same network configuration for training and testing are used in these works.

The discrepancy between the prediction accuracy in the testing set and the forecasted probability by multi-class logistic regression in neural network has been reported earlier in Guo et al. (2017). There are many other works on uncertainty quantification of prediction; for a comprehensive review, see Abdar et al. (2021) and Gawlikowski et al. (2023)

## 3 METHOD

### 3.1 LIMITATION OF CURRENT CNN TRAINING FOR POST-PREDICTION EVALUATION OF MODEL ADEQUACY

Complex models such as CNN involve more parameters than cases for training. It would be impractical to performance a global model criticism without compartmentalizing the model into smaller components. Consider a model taking the form $\mathbf{o} = g(\mathbf{h}, \theta_1), \mathbf{h} = f(\mathbf{x}, \theta_2)$ where the dimension of $\theta_1$ is of a smaller order than the training sample size. The model criticism is focused on the first component. In supervised learning, training and testing often follow the same model. The task of training is to minimize certain loss functions that measure the inaccuracy of prediction. After the training is completed, the prediction step is easy :

(1) Set the parameter values to the trained values, $\theta_1 = \hat{\theta}_1, \theta_2 = \hat{\theta}_2$.

(2) Input a test case $\mathbf{x}^*$ of interest to the model and receive the output $\mathbf{o}^* = g(\mathbf{h}^*, \hat{\theta}_1), \mathbf{h}^* = f(\mathbf{x}^*, \hat{\theta}_2)$.

However, this practice leaves very little room for investigating the potential model inadequacy that may harm the quality of prediction for this particular input case. After prediction, all information about $\mathbf{x}^*$ has been trimmed down to $\mathbf{o}^*$ and $\mathbf{h}^*$. Because $\mathbf{o}^*$ and $\mathbf{h}^*$ are obtained under the framework of f and g, it is self-limiting to rely only on them for model criticism. For example, if the prediction for $\mathbf{x}^*$ fails because other features in $\mathbf{x}^*$ have escaped the scrutiny of the prediction model, then there are no ways to find out. Such features may be unique to a small subpopulation of the class that $\mathbf{x}^*$ belongs to and yet are also present in members of other classes. To alleviate the limitation of CNN training, it is imperative to separate the training model configuration from the prediction model configuration.

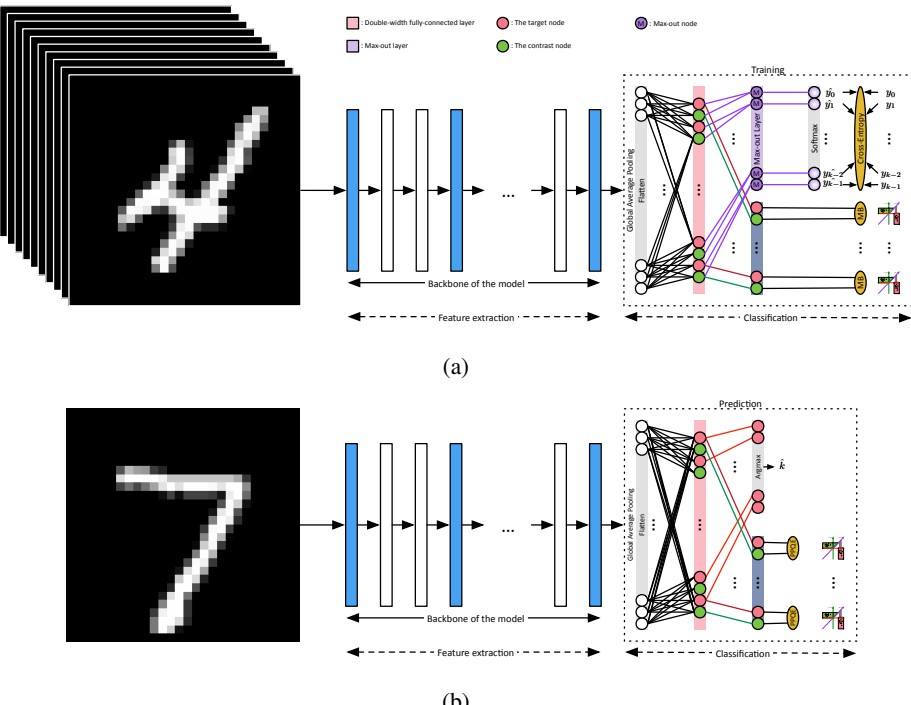

Figure 2: An illustration of (a) the training configuration and (b) the prediction configuration.

### 3.2 THE AUGMENTATION OF PREDICTION MODEL CONFIGURATION IN TRAINING FOR PARTIAL MODEL CRITICISM

We use "e" and "r" to differentiate testing and training. We augment the model for testing: $\mathbf{o_e} = g_e(\mathbf{h}, \theta_1), \mathbf{h} = f(\mathbf{x}, \theta_2)$, to a large model for training: $\mathbf{o_r} = g_r(\mathbf{h}, \theta_1, \gamma), \mathbf{h} = f(\mathbf{x}, \theta_2)$. Certain relationship between $g_e$ and $g_r$ must be established so that the training model can be viewed as an augmentation of the testing model. Consider the situation that $g_e$ can be broken down into the composite of two functions: $g_e = g_0(g_1(\mathbf{h}, \theta_1))$. Then an augmented model may take the form of $g_r = g_0(g_2(g_1(\mathbf{h}, \theta_1), g_1(\mathbf{h}, \gamma)))$. After the training is completed, we obtain optimal parameter values $\hat{\theta}_1, \hat{\theta}_2, \hat{\gamma}$. The prediction will proceed in the following way:

(1) Setup the parameter values in the testing model: $\theta_1 = \hat{\theta}_1, \theta_2 = \hat{\theta}_2$

(2) Input a test case $\mathbf{x}^*$ of interest to the testing model and receive the output $\mathbf{o}^* = g_0(g_1(\mathbf{h}^*, \hat{\theta}_1), \mathbf{h}^* = f(\mathbf{x}^*, \hat{\theta}_2)$ for prediction.

(3) For model criticism, input $\mathbf{x}^*$ to the training model and receive $g_2(\mathbf{h}^*, \hat{\gamma})$.

(4) Use $g_1(\mathbf{h}^*, \hat{\theta}_1)$ and $g_2(\mathbf{h}^*, \hat{\gamma})$ to perform post-prediction diagnosis.

For Deep CNN, take $\theta_1$ as $\phi$ and recognize $\mathbf{t} = g_1(\mathbf{h}, \theta_1)$ and $g_0(\mathbf{t}) = \text{softmax}(\mathbf{t})$. For $g_2$, we shall consider the bivariate maxout function, $g_2(a, b) = \max\{a, b\}$. Figure 2 is a blueprint of the network configuration.

### 3.3 T CHANNEL AND C CHANNEL: THE DESIGN PRINCIPLE OF YO-YO MOLDING PLATE.

There are K pairs of T versus C channels in Figure 2(a). For pair k, the T channel is designed to capture the primary feature of class k (the target group) and the C channel is designed to capture main features for members of other classes (the contrastive group). The following four rules are required to differentiate the roles of T channels and C channels if training is completed perfectly. Consider the range of values $(t, c)$ of T and C channels.

**1.** $T^t > C^t$, $C^c > T^c$: For members from the target group, $t > c$; for members form the contrastive group, $t < c$.

**2.** $T^t_{min} > T^c_{max}$: The minimal value of the T channel for members from the target group $T^t_{min} = \min\{t|target\ group\}$ is larger than the maximal value of the T channel for the contrastive group $T^c_{max} = \max\{t|contrastive\ group\}$.

**3.** $C^c_{min} > C^t_{max}$: The minimal value of the C channel for the contrastive group $C^c_{min} = \min\{c|contrative\ group\}$ is larger than the maximal value of the C channel for the target group $C^t_{max} = \max\{c|target\ group\}$.

**4.** $T^t_{min} > C^c_{max}$: The minimal value of the T channel of the target group $T^t_{min} = \min\{t|target\ group\}$ is larger than the maximal value of the C channel of contrast group $C^c_{min} = \max\{c|contrative\ group\}$.

Following these rules, a molding plate, named Yo-Yo, is introduced to represent the ideal pattern of features captured through the two channels after training for every class. An illustration of Yo-Yo plate is shown in Figure. 3.

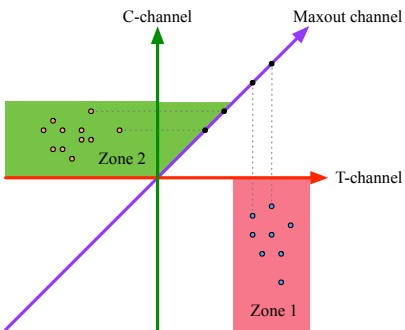

Figure 3: The Yo-Yo Plate

### 3.4 MOULDING BONUS AND ALGORITHM OF TRAINING

To coerce a batch of training samples into the Yo-Yo molding during training, we specify a bonus function to award the rule-obeying.

$$MB = \frac{1}{N}\frac{1}{K}\sum_{i=1}^{N}\sum_{k=1}^{K} PB(t_{ik}, c_{ik}, y_{ik}; \mu_1^T, \mu_1^C, \mu_0^T, \mu_0^C, \omega_1^T, \omega_1^C, \omega_0^T, \omega_0^C), \qquad (1)$$

$$= \frac{1}{N}\frac{1}{K}\sum_{i=1}^{N}\sum_{k=1}^{K} \lambda_{y_{ik}}[SB(t_{ik}; \mu_{y_{ik}}^T, \omega_{y_{ik}}^T) + SB(c_{ik}; \mu_{y_{ik}}^C, \omega_{y_{ik}}^C)], \qquad (2)$$

$$SB(x; \mu, \omega) = \log(\sigma(x - (\mu - \omega/2))) + \log(\sigma(-(x - (\mu + \omega/2)))), \qquad (3)$$

where SB is the bonus of a single channel, PB is the bonus of a pair of (T,C) channels, and MB is the total molding bonus. Here $t_i$ and $c_i$ are the values of T channel and C channel of an input $x_i$ with

1-hot class membership vector $y_i$. The $(\mu_1^T, \mu_1^C, \mu_0^T, \mu_0^C)$ are the designed centers of T channel and C channel for target and contrastive groups respectively. The $(\omega_1^T, \omega_1^C, \omega_0^T, \omega_0^C)$ are the designed distribution width. The $\lambda_1$ and $\lambda_0$ is the weights designed to balance sample sizes of target group and contrastive group. In addition to the bonus function, we also keep the cross-entropy

$$CE = -\frac{1}{N}\sum_{i=1}^{N}\sum_{k=1}^{K} y_{ik} * \log(\hat{y}_{ik}) \tag{4}$$

where $y_{ik} = 1$ if the label of $i_{th}$ case is k and $y_{ik} = 0$ otherwise. Our goal is to maximize bonus while minimizing CE. Putting together, we shall minimize the final loss function

$$Loss = CE - MB \tag{5}$$

### 3.5 Post-prediction quality evaluation

For a given input $\mathbf{x}^*$, we use T channel for class membership prediction $\hat{k} = \arg\max(\mathbf{t}^*)$. We also receive K pairs of Yo-Yo scores $(t_i^*, c_i^*), i = 1, \cdots, K$. Comparing each Yo-Yo score pair with the Yo-Yo plate rule, we also receive a rule-violation indicator $(v_1, \cdots, v_K)$ where $v_i = 1$ if $(t_i^*, c_i^*)$ does not fall inside of the proper zone setup by Yo-Yo plate. Since the predicted class is $\hat{k}$, for plate $i = \hat{k}$, must fall in Zone 1 (the target group zone), not Zone 2 (the constrative group zone). That is: (i) $v_{\hat{k}} = 0$ if $(t_{\hat{k}}^*, c_{\hat{k}}^*) \in Zone\ 1$, $v_{\hat{k}} = 1$ otherwise, and (ii) $v_i = 0$ if $(t_i^*, c_i^*) \in Zone\ 2$, $v_i = 1$ otherwise for $i \neq \hat{k}$.

**Two-category grading.** High quality prediction (HQP): no zone violations are found. Low quality prediction(LQP): one or more zone-violations are found.

**Numerical grading by distance inside the target zone.** For a prediction $\hat{k}$ with HQP, we can assign a finer numerical quality score by examining the Yo-Yo plate of the predicted class $\hat{k}$ more closely. We measure how far away from the side of zone boundary facing the constructive zone, the point $(t_{\hat{k}}^*, c_{\hat{k}}^*)$ lands. The closer the distance, the lower the quality relatively. Let $d_*$ be the obtained distance. It is easier to interpret the distance score by comparing with a reference distribution. We use the distances obtained by the training dataset as the reference, and report the quantile as an estimated $p$-value; namely $\hat{p} = \hat{F}^{-1}(d^*)$, where $\hat{F}$ is the empirical distribution of $d_n, n = 1, \cdots, N$, $N$ is the total number of training datapoints, and $d_n$ is a distance score.

**Graphical output for detailed diagnostics.** For a batch of B input queries, $\mathbf{x_1^*}, \cdots, \mathbf{x_B^*}$, the output Yo-Yo scores are displaced in the K Yo-Yo plates. The Yo-Yo plates use the data clouds of the reference distance distribution as the background. Figure 4 is an example of Yo-Yo plates for B=4. The target classes for Yo-Yo plates are 1=airplane, 2=automobile, 3=bird, 4=cat, 5=deer, 6=dog, 7=frog, 8=horse, 9=ship, 10=truck. CIFAR-10 is used in the illustration.

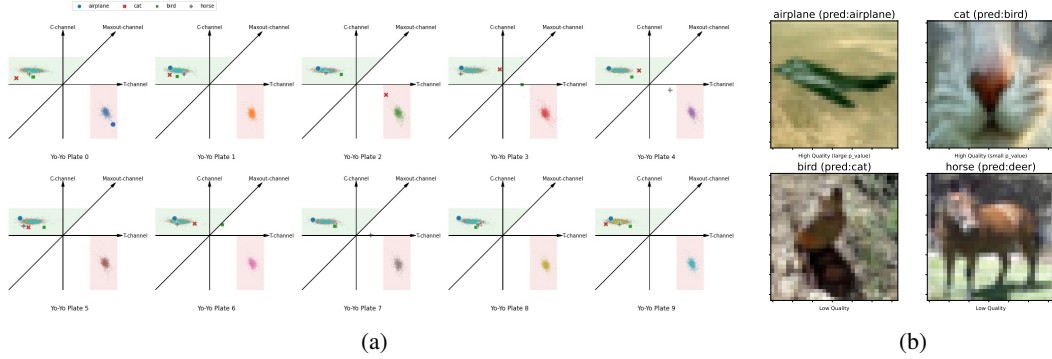

Figure 4: (a) The Yo-Yo plates; (b) Images of 4 test examples.

## 4    EXPERIMENTS

Two popular CNN architecture, ResNet (He et al., 2016) and VGG (Simonyan & Zisserman, 2014), are chosen as our backbones. All experiments were performed on a single NVIDIA Quadro RTX 8000 GPU with an 1.5T-memory server using TensorFlow.

**Datasets.**    Five image datasets, MNIST (Deng (2012)), Fashion-MNIST (Xiao et al. (2017)), CIFAR-10 (Krizhevsky et al., 2009), CIFAR-100, STL-10 (Coates et al. (2011)) are used to implement our method. Before training procedure, we calculate the mean and standard deviation of the training set and use them to normalize the training and testing set. The normalization is performed individually in each R, G, and B channel if the dataset is color coded. The image size is (28, 28, 1) for the black-white MNIST and Fashion MNIST. For the color-coded images, CIFAR-10 and CIFAR-100 have (32, 32, 3) while STL-10 has (96, 96, 3).

**Backbone CNN architecture.**    VGG16 is adopted for MNIST and Fashion-MNIST datasets. The VGG16's capability is enable to handle these two datasets because of the relatively lower complexity of these two datasets. For CIFAR-1 and CIFAR-100, we use ResNet50 as the backbone with input size (224, 224). The number of output image features extracted by these two backbones are 512 and 2048 respectively.

**Training the Yo-Yo augmented CNN.** We adopt the pre-trained parameters with ImageNet as the seeding in training the augmented CNN. Figure 2(a) shows the configuration of our Yo-Yo training model. Adam (Kingma & Ba, 2014) optimizer with default $\beta_1$ and $\beta_2$. During training, the batch size is 32 for MNIST and Fashion-MNIST, and is 200 for CIFAR-10 and CIFAR-100. The DataAugmentation and ReduceLROnPlateau strategies are applied. For the Yo-Yo molding bonus functions, the parameters values are that $(\mu_1^T, \mu_1^C, \mu_0^T, \mu_0^C) = (15, -10, -10, 5)$ and $(\omega_1^T, \omega_1^C, \omega_0^T, \omega_0^C) = (10, 20, 20, 10)$.

The following figure, Figure 5, shows a typical datapoint pattern of Yo-Yo plates for the first epoch of training and the last epoch. For the first epoch, many training cases are landing outside of the target. At the end of zone, all cases approximately land inside the target zone. CIFAR-10 is used in the illustration.

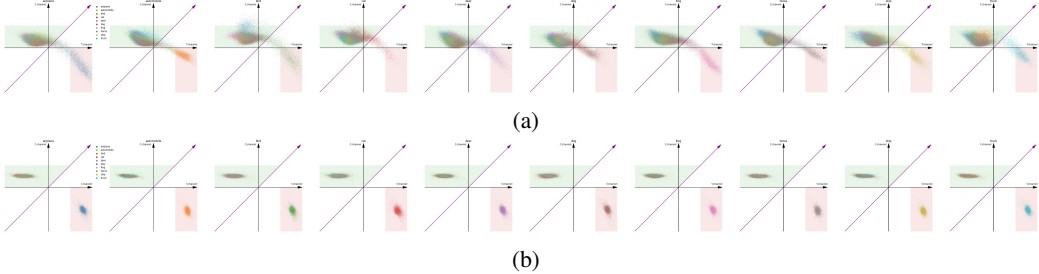

(a)

(b)

Figure 5: (a) The first epoch; (b) The end epoch.

**Results.**    The overall prediction accuracy of our method and that of the backbone CNN are compared, and the results are shown in Table. 1. Our method performs as well as the backbone in four datasets and outperform the backbone in one dataset. Furthermore, the high quality predictions have achieved a much higher level of accuracy than the lower quality predictions. The error rate ratio, $\frac{1-\text{accuracy of LQP}}{1-\text{accuracy of HQP}}$, ranges between 3.90 (STL-10) to 50.73 (MNIST).

## 5    DISCUSSION

### 5.1    COMPRESSED CLASS-LABEL REPRESENTATION

For CIFAR-100, although the precision for HQP is high (98.77%), out of 10000 test cases, only 4146 cases are HQP. This is due to the increased number of Yo-Yo plates to monitor. To overcome this limitation, we used a compressed representation to reduce the Yo-Yo plates from 100 to 15. Instead of using one-hot vector (100 dimension) to represent a class, we use a two-hot vector (15

Table 1: Comparison Table

| Dataset | MNIST | Fashion MNSIT | CIFAR-10 | CIFAR-100 | CIFAR-100 (k-hot) | STL-10 |
|---|---|---|---|---|---|---|
| Backbone | VGG16 | VGG16 | ResNet50 | ResNet50 | | ResNet50 |
| Image size | (28,28,1) | (28,28,1) | (32,32,3) | (32,32,3) | | (96,96,3) |
| Size of Training set | 6000*10 | 6000*10 | 5000*10 | 500*100 | | 500*10 |
| Size of Testing set | 1000*10 | 1000*10 | 1000*10 | 100*100 | | 800*10 |
| Accuracy of backbone model(%) | 99.65 | 95.02 | 96.57 | 83.45 | - | 65.00 |
| Accuracy of Yo-Yo model (%) | 99.43 | 95.12 | 96.52 | 83.55 | 80.50 | 73.75 |
| Accuracy of HQP (%) | 99.77 (9940) | 97.25 (9349) | 98.71 (9208) | 98.77 (4146) | 94.42 (6956) | 86.54 (5387) |
| Accuracy of LQP (%) | 88.33 (60) | 60.83 (651) | 73.11 (792) | 72.60 (5854) | 48.62 (3044) | 47.46 (2613) |
| error rate ratio | 50.73 | 14.24 | 20.84 | 22.28 | 9.21 | 3.90 |

dimension) to represent a class and employ cross-entropy loss function between the softmax of the fifteen logits and the two-hot vector. Imagine there are 15 different bags. Each class is connected to the 15 bags with two edges. So each class is represented by a 0-1 vector of dimension 15, with 1 appearing in exactly two places. Since there are $\binom{15}{2} = 105$ two-hot vectors available, there are five unused two-hot vectors. This causes a slight unevenness in the number of classes connected to each bag. To resolve this issue, we employ an idea form PPS (probability proportion to size) sampling in the survey sampling literature to modify the cross entropy. See Appendix for details. As a result, the number of HQP is increased to 6957.

## 5.2 THE DISTRIBUTION OF EXTRACTED IMAGE FEATURES

Compared to the backbone CNN, the augmented Yo-Yo network configuration also induces a great change on the extracted image feature layer $\mathbf{h}$. To visualize the change, we conduct a principal component analysis on $\mathbf{h}$ for the data yielded by the backbone and Yo-Yo models. The results for CIFAR-10 is given in Figure 6. The lower-left halt of the scatterplot matrix is the result of Yo-Yo model and upper-right half is from the backbone model.

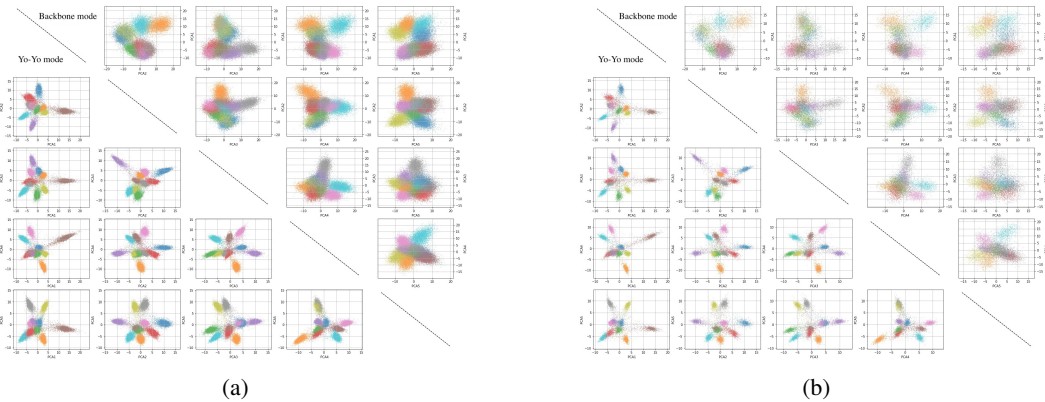

(a)          (b)

Figure 6: Scatterplot matrix for the first five PCA directions (a) Training set of CIFAR-10; (b) Testing set of CIFAR-10.

## 5.3 MOLDING BONUS FUNCTION

The molding bonus function in our implementation is plotted in Figure 7. The center and width are set to 10 and 20 respectively. The function reaches the highest point at x = 10. It should be noted that the gradient of the bonus curve stays approximately zero in a window centered at 10.

There are many other choices to serve the same purpose that can explored function. In principle, any hill-like convex function with flat hill top can be a candidate.

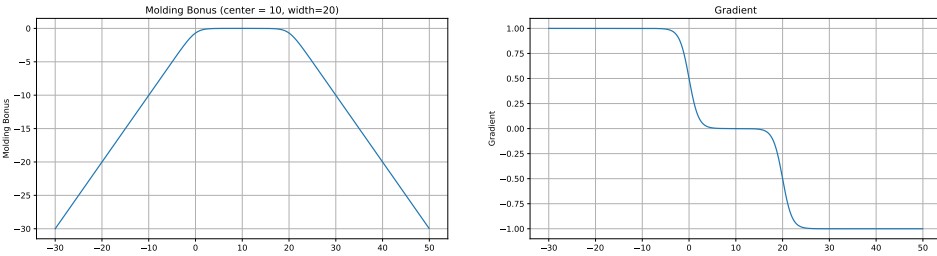

Figure 7: The curves of the molding bonus function (left) and its gradient (right)

### 5.4    P-VALUE FOR INSIDE-PLATE DISTANCE

The reference distribution we used is based on the entire set of training set. Alternatively, we can use the split-sample strategy to perform training on a proper training set and use the remaining cases in the training set for generating a reference distribution. We have tried this strategy but found little difference from the curve generated by using the whole training set; see Appendix.

### 5.5    LOW QUALITY PREDICTION

For some applications including active learning, the LQP cases will receive more attention than the HQP cases. The Yo-Yo plates can be examined in detail to prioritize the order of selecting the cases that may receive most benefit from the teacher. For instance, in medical application, the bioassay for determining the exact class membership may be more expansive than imaging. Upon receiving a batch of image-based prediction with mixed quality, one may want to send the LQP for bioassay. Suppose the assays should be conducted separately for determining different classes. Then a choice has to be made which cases of LQP should be sent out first and for what assays. For example, one may start with LQP with one or two Yo-Yo plate violations and send out for assaying the target classes with Yo-Yo plate violations. Such decisions should be explored to meet the purpose of each application.

## 6    CONCLUSION

This work is concerned about multiple aspects on the quality of a prediction from a user's perspective after a prediction is made. Three sources of prediction uncertainty are addressed. Post-prediction confidence training aims at offering users a wider spectrum of prediction quality summary useful for building up their confidence. One distinctive feature of our approach is the required network architecture enhancement in order to open up the angle for model criticism. This is enabled by a maxout Yo-Yo plate design. Both categorical quality grading and numerical grading are offered after prediction, along with the graphical out of Yo-Yo plates for more detailed quality inspection. In addition, the inside zone distance can be used as a conformance metric in applying inductive CP for addressing the uncertainty due to the population mismatching between training and testing. Our efforts of imposing logic lucidity on the Yo-Yo plate design is in line with the recent demand on explainable AI.

## 7    REPRODUCIBILITY STATEMENT

In this paper, we use the ImageNet pre-trained parameter and datasets providing from TensorFlow 2.0 libraries. We also provide the customized loss function, including molding bonus function in Appendix. In the experiment section, we provide the values of $\mu$ and $\omega$ used in our experiments for the reproducibility. The results are reproducible based on the provided information.

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

# A APPENDIX

## A.1 REFERENCE DISTANCE DISTRIBUTION

We provide the curves of cumulative distribution function of the training, validation and testing set. We randomly select the images from training set as the validation set. Its size is equal to the size of testing set. It should be noted that the curve of the testing set is similar to the validation set. CIFAR-10 is the dataset used in the illustration (Figure 8).

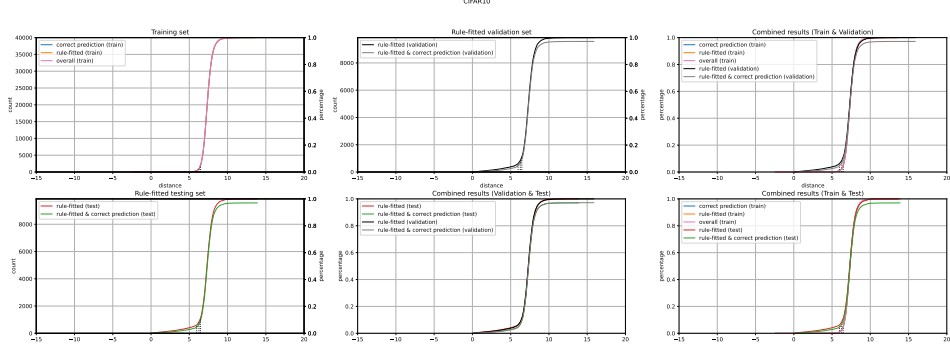

Figure 8: The curves of cumulative distribution function, including training, HQP validation set and HQP testing set.

## A.2 COMPRESSED CLASS-LABEL REPRESENTATION

$$\hat{Y}_q = P_i * P_j * \left( \frac{1}{1 - P_i} + \frac{1}{1 - P_j} \right) \tag{6}$$

$$P_i = \frac{e^{m_i}}{\sum_{s=0}^{14} e^{m_s}} \tag{7}$$

The $\hat{Y}_q$ is the output value of the q-th class connected with the i-th and j-th nodes of the maxout layer. The architecture of PPS module is shown in the Figure 9. The configuration of CIFAR-100 dataset is the example used in the illustration.

## A.3 YO-YO PLATES

In additional to CIFAR-10 result, we provide the Yo-Yo plates of the end epoch of other datasets, MNIST, Fashion MNIST and 10 examples of CIFAR-100. It should be noted that the cases approximately land inside the target zone.

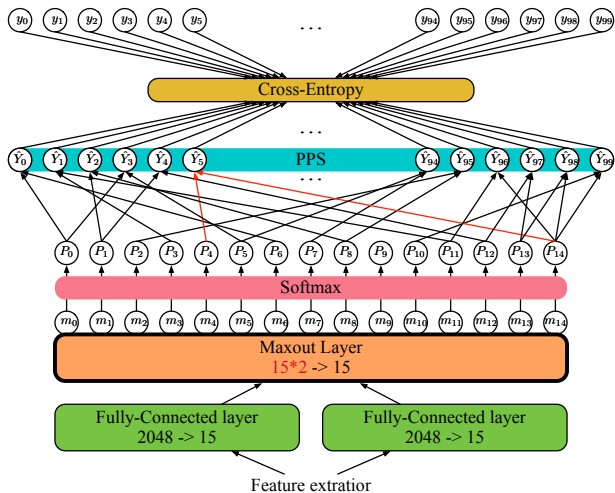

Figure 9: The architecture of the classification part in the PPS-based model

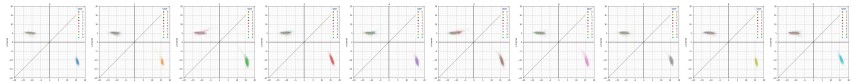

Figure 10: The end epoch of the training set of MNIST

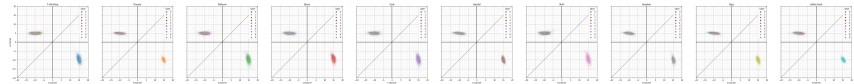

Figure 11: The end epoch of the training set of Fashion MNIST

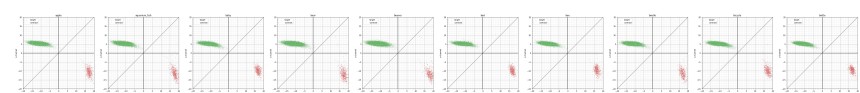

Figure 12: The end epoch of the training set of the 10 Yo-Yo plats of the CIFAR-100

## A.4 SOURCE CODE

In appendix, we provide the some snapshots of the key source codes including the molding bonus function. They are implemented with TensorFlow 2.0 version. We also can provide the complete through the anonymized link for downloading

```python
def custom_loss_mb(onehot_depth = 10, weight = 10):

    contrast_xmin, contrast_xmax, focus_xmin, focus_xmax = [-20,  0,  10, 20]
    contrast_ymin, contrast_ymax, focus_ymin, focus_ymax = [  0, 10, -20,  0]

    # Create a loss function that adds the MSE loss to the mean of all squared activations of a specific layer
    def loss(y_true, y_pred):
        y_true = y_true[...,0]
        y_pred_0 = y_pred[..., 0:onehot_depth]

        y_true_onehot   = tf.one_hot(y_true, depth = onehot_depth)

        output_split_0 = y_pred[..., onehot_depth  ::2]
        output_split_1 = y_pred[..., onehot_depth+1::2]

        # Edited by (2023.02.21)

        output_sigmoid_0 = tf.where(y_true_onehot == tf.zeros_like(y_true_onehot), -tf.math.log(tf.clip_by_value(tf.math.sigmoid(  output_split_0 - contrast_ymin) ,1e-30,1.0))\
                                                                                   -tf.math.log(tf.clip_by_value(tf.math.sigmoid(-(output_split_0 - contrast_ymax)),1e-30,1.0)),
                                                                                   weight*(-tf.math.log(tf.clip_by_value(tf.math.sigmoid(  output_split_0 - focus_ymin) ,1e-30,1.0))
                                                                                   -tf.math.log(tf.clip_by_value(tf.math.sigmoid(-(output_split_0 - focus_ymax)),1e-30,1.0))))

        output_sigmoid_1 = tf.where(y_true_onehot == tf.zeros_like(y_true_onehot), -tf.math.log(tf.clip_by_value(tf.math.sigmoid(  output_split_1 - contrast_xmin) ,1e-30,1.0))\
                                                                                   -tf.math.log(tf.clip_by_value(tf.math.sigmoid(-(output_split_1 - contrast_xmax)),1e-30,1.0)),\
                                                                                   weight*(-tf.math.log(tf.clip_by_value(tf.math.sigmoid(  output_split_1 - focus_xmin) ,1e-30,1.0))\
                                                                                   -tf.math.log(tf.clip_by_value(tf.math.sigmoid(-(output_split_1 - focus_xmax)),1e-30,1.0))))

        loss_ce = tf.keras.losses.SparseCategoricalCrossentropy(from_logits=True)(y_true, y_pred_0)

        return loss_ce + tf.math.reduce_mean(output_sigmoid_1)+tf.math.reduce_mean(output_sigmoid_0)

    # Return a function
    return loss
```

Figure 13: The source code of final loss function, including molding bonus and cross-entropy.

```python
def model_example(input_shape, o_units = 10, mode = 0):
    backbone = ResNet50(include_top = False, weights = "imagenet", input_shape = (224, 224, 3))

    input_1 = tf.keras.layers.Input(shape=input_shape, dtype=np.float32)
    output = tf.keras.layers.UpSampling2D(size=(7, 7),interpolation='bicubic')(input_1)

    output = backbone(output)

    output = tf.keras.layers.GlobalAveragePooling2D()(output)

    if mode == 0:
        output = tf.keras.layers.Dense(units = o_units)(output)

        model = tf.keras.Model(inputs=[input_1], outputs=output)
    else :
        output = tf.keras.layers.Dense(units = o_units*2)(output)

        output_2w = tf.keras.layers.Reshape((o_units, 2))(output)

        output_class = tf.keras.layers.MaxPool1D(pool_size=2, data_format="channels_first")(output_2w)
        output_class = tf.keras.layers.Flatten()(output_class)
        output_concat = tf.keras.layers.concatenate([output_class, output], axis=1)

        model = tf.keras.Model(inputs=[input_1], outputs=[output_concat])

    return model
```

Figure 14: The source code of the model, including Yo-Yo and backbone models.

