# OpenReview forum: "Post-prediction confidence training complements supervised learning"
_ICLR.cc/2024/Conference — ICLR 2024 Conference Withdrawn Submission_

### Official Review · Reviewer_2myu · 2023-10-26

**Soundness:** 2 fair
**Presentation:** 1 poor
**Contribution:** 2 fair
**Rating:** 1
**Confidence:** 4

**Summary:**

This paper proposes a post-prediction diagnosis method, which involves training with a penalty that encourages learning class-wise feature and contrastive feature (T and C channel), which can then be plotted and decide whether a prediction a high/low quality. If the features align with the prediction, this is high quality, otherwise low quality.

**Strengths:**

1. Examining the consistency of the feature as a confidence measure is intuitive and a common approach to similar problems.

**Weaknesses:**

1. This paper is missing a lot of important related works and does not seem to follow recent research. For example, "However, to the best of our knowledge, successful applications of these methods, including inductive conformal prediction, to deep CNN on large image datasets have not been reported." This is clearly wrong... as early as in 2020 there was https://openreview.net/forum?id=eNdiU_DbM9 on ImageNet.
	a. The Related Work is mostly Conformal Prediction when this is clearly about model confidence/classification with rejection.

2. Insufficient experiments: No baseline. A naive baseline would be if we just set some threshold on the maximum class probability we can also separate HQP and LQP. Also the paper suggested "large image dataset" yet only CIFAR-100 is used (and it seems like there were some difficulty applying the proposed method to 100 classes as well).


3. Writing is very bad - not just in terms of presentation but the entire draft is incoherent. For example,
	a. captions are uninformative;
	b. The part about Yo-Yo plate is extremely terse despite the importance. Also, is point 1 in 3.3 a typo? I don't see "t>c" (lower or upper case) in the two equations. Similar typo in point 4 as well for $C_min$
	c. No introduction leading to section 3.3 at all.
	d. the 1st contribution is "pinpointing three sources of uncertainty" including model imperfectness, yet "imperfect" never appeared after the introduction section.

In general, I could see how this paper could be improved to an acceptable paper, but the current form does not look like a complete version actually.

**Questions:**

1. What message does Fig 4 convey? I think I understand but there's little explanation and I'd like to confirm. Also the font is unreadable, and "Plate 0" should be replaced with "Airplane" (also I assume 0=airplane as the index does not match).

2. What does 1b convey in this paper? There's no mention of calibration in this paper.

---

### Official Review · Reviewer_c9j6 · 2023-10-29

**Soundness:** 1 poor
**Presentation:** 2 fair
**Contribution:** 1 poor
**Rating:** 3
**Confidence:** 4

**Summary:**

The paper discusses the problem of getting confidence estimation of a CNN, where the usual softmax output does not represent the true probability, resulting in calibration error. The paper then proposes a method of augmenting the baseline CNN and use contrastive learning to obtain the quality of a prediction.

**Strengths:**

The paper’s idea seem interesting and the paper is relatively easy to read.

**Weaknesses:**

**Overall comment**

I had a hard time understanding what is being discussed in the paper. The authors seem to have missed a lot of relevant literature/even the problem setup, and have weak evaluations of their claims.

**Do not cite/compare to calibration literature**

>  But even a prediction with 95% probability concentrated on one class may still turn out wrong many times more often than the anticipated rate of 5%.

> Among all test cases falling between .95 and .96, less than 60% cases are correctly predicted. Compared to the anticipated error rate of 5%, the actual error rate is more than 8 times higher.

This is the calibration error problem. Calibration is defined as the difference between true probability given a model’s prediction, and the model’s predicted probability. There exists numerous prior work in trying to solve the calibration problem, I do not see any of them cited by this paper: Platt scaling [1], Isotonic Regression [2], scaling-binning calibrator [3] to name a few.

No comparison/discussion of any of these methods is given.

**Poor experimental evaluation**

The quality and depth of experimental evaluation is poor due to the lack of appropriate baselines, lack of ablation studies. The experiments only use very small image datasets, and no experiments are done on large datasets like ImageNet [4] / WILDS [5].

**No relationship between claims in introduction and experiments done, claims are not formal**

> We pinpoint three sources that may cause prediction uncertainty in deep CNN: distributional non-conformability, population heterogeneity, model imperfectness

Where is this pinpointing happening? What is the definition of each of these terms? E.g., what is model imperfectness here? Which experiments are done to establish these three sources?

**Typos**

The paper also has some typos:
1. Sever -> Severe (page 1)
2. CIFAR1 -> CIFAR10 (page 7)

etc.

**Questions:**

1. (**Compressed class label representation**) How is the class compression being done? There is an official superclass clustering available for CIFAR-100, see [6]
2. Also what is the final confidence value you get for a prediction? How is that number compared to the baseline CNN? Could you provide a reliability diagram [7][8] of your method? Also there are other metrics to measure calibration error, like ECE [3], could you report some of these?

[1] Probabilistic outputs for support vector machines and comparisons to regularized likelihood methods. https://home.cs.colorado.edu/~mozer/Teaching/syllabi/6622/papers/Platt1999.pdf

[2] Transforming classifier scores into accurate multiclass probability estimates, https://dl.acm.org/doi/10.1145/775047.775151

[3] Verified Uncertainty Calibration, https://arxiv.org/abs/1706.04599

[4] ImageNet: A large-scale hierarchical image database, https://ieeexplore.ieee.org/document/5206848/authors#authors

[5] WILDS: A Benchmark of in-the-Wild Distribution Shifts, https://arxiv.org/abs/2012.07421

[6] Coarse CIFAR-100: https://github.com/ryanchankh/cifar100coarse

[7] The comparison and evaluation of forecasters. Journal of the Royal Statistical Society. Series D (The Statistician), 32:12–22, 1983.

[8]  Increasing the reliability of reliability diagrams. Weather and Forecasting, 22(3):651–661, 2007

---

### Official Review · Reviewer_xsnZ · 2023-10-30

**Soundness:** 2 fair
**Presentation:** 1 poor
**Contribution:** 2 fair
**Rating:** 3
**Confidence:** 3

**Summary:**

The paper deals with the important issue of prediction uncertainty in Convolutional Neural Networks (CNNs). It introduces a novel method called post-prediction confidence training (PPCT) to guide users in distinguishing between high and low-quality predictions by considering three sources of uncertainty: distributional non-conformability, population heterogeneity, and model imperfectness.

The authors rightly point out that while the overall accuracy of a machine learning model is a critical concern for developers, the individual users are more interested in the trustworthiness of specific predictions. The paper delves into the gaps in confidence measures and presents a novel approach, PPCT, to address these concerns. Notably, it introduces modifications to the network's training architecture and provides a comprehensive post-prediction diagnostic summary for users.

**Strengths:**

The authors introduce a new method (PPCT) that focuses on post-prediction confidence, a significant and under-researched issue in the deep learning community.
The paper does an excellent job of identifying the various sources of prediction uncertainty, making it a comprehensive study on the topic.
The proposed modifications to the network architecture, dual-channel representation, and the introduction of the Yo-Yo molding plate, are interesting and innovative.
The post-prediction diagnostic summary offers real practical value, enabling users to evaluate predictions from multiple perspectives.

**Weaknesses:**

The paper doesn't adequately compare PPCT with existing conformal inference methods, which would be crucial for establishing its superiority or identifying its unique benefits. Like Uncertainty sets for image classifiers using conformal prediction
by Angelopoulos et al. (2020).
The methods developed lack rigorous theoretical guarantees, making it challenging to ascertain their reliability and robustness while conformal inference methods provide precise guarantees that hold in finite sample and under minimum assumptions.
The paper doesn't seem fully updated on the latest advancements in conformal inference literature, which is crucial for a comprehensive review of the subject. For example, besides using conformal inference as post processing step, there have been developments in tuning hyperparameters using conformal inference, like in Optimizing Hyperparameters with Conformal Quantile Regression by Salinas et al. (ICML 2023).
The numerous grammatical errors and typos make the paper difficult to read and comprehend in some sections, detracting from its quality and clarity.
The list of weaknesses provided by the reviewer seems incomplete, and a more thorough critique would be beneficial.

The authors should conduct extensive comparisons with existing conformal inference methods, providing both quantitative and qualitative analyses.
 It would be valuable if the authors can offer some theoretical guarantees or bounds for their method, even if under certain conditions or assumptions or at least carry out more extensive experiments on additional datasets highlighting the nuances of the method and providing deeper intuition on the varying degree of effectiveness of the method.
 A more comprehensive literature review on conformal inference would enrich the paper's content and context.
The paper should be thoroughly proofread to eliminate grammatical and typographical errors, ensuring clearer communication of ideas.

**Questions:**

How does this work differentiate or improve upon previously published methods on the same topic? A clearer distinction could help underline the significance of this study.

Have the authors considered testing their methodology across other datasets or domains? This would help gauge the robustness and versatility of their approach.

Some sections of the paper could benefit from visual representations or flow diagrams to help the reader grasp the concept more intuitively.

---

### Official Review · Reviewer_9DJD · 2023-11-01

**Soundness:** 3 good
**Presentation:** 2 fair
**Contribution:** 2 fair
**Rating:** 3
**Confidence:** 3

**Summary:**

In this work, the authors focus on analyzing the causes of prediction uncertainty in the context of supervised learning, including distribution shift, subpopulation shift across classes, and imperfect prediction model. While previous works focused on distribution shift, the authors propose a new approach called post-prediction confidence training to explore the other two reasons. They also suggest a blueprint, which uses T channel and C channel in CNN configuration. Experiments on several image datasets are conducted to show the effectiveness of the proposed method.

**Strengths:**

1. The problem explored in this work is crucial to the development of modern deep learning. Understanding the quality of predictions is a critical task in real-world applications.

2. The method provided in this work can address three sources of prediction uncertainty. The idea is novel as previous method focused on distribution shift only.

**Weaknesses:**

1. The writing can be improved. The authors directly describe their method in the main part, instead of introducing the problem setting. This might increase the difficulty of readers to touch the background.

2. The motivation of the proposed method is not clear. Although the authors introduce the limitation of CNN training for post-prediction evaluation in subsection 3.1, the description is not clear so that the reviewer is still confused about what the limitation is.

3. There are too many hyperparameters in the proposed method. As introduced in Subsection 3.4, There are at least ten hyperparameters that are chosen by human, which I think is impracticable.

4. It can be better if the authors can provide empirical evidences by large-scale datasets, like Imagenet.


Typo: Section2 "Matiz & Barner (2019) applied ***IPC*** to active learning.", should be "ICP"

**Questions:**

1. In subsection3.1, The authors introduce the problem for complex models and list CNN as an example. I am confused why the authors describe the limitation as an issue of CNN training, instead of all complex models. Can we implement the proposed method for other network architectures, like transformer?